# The Inferential Binding Sites of GCGR for Small Molecules Using Protein Dynamic Conformations and Crystal Structures

**DOI:** 10.3390/ijms25158389

**Published:** 2024-08-01

**Authors:** Mengru Wang, Xulei Fu, Limin Du, Fan Shi, Zichong Huang, Linlin Yang

**Affiliations:** 1Department of Pharmacology, School of Basic Medical Sciences, Zhengzhou University, Zhengzhou 450001, China; mrwang0319@163.com (M.W.); 15128077927@163.com (X.F.); dminmin99@163.com (L.D.); shi008769@gs.zzu.edu.cn (F.S.); 2Green Catalysis Center, College of Chemistry, Zhengzhou University, Zhengzhou 450001, China; hzc2456048517@gs.zzu.edu.cn

**Keywords:** GCGR, small molecules, dynamic conformations, binding sites

## Abstract

Glucagon receptor (GCGR) is a class B1 G-protein-coupled receptor that plays a crucial role in maintaining human blood glucose homeostasis and is a significant target for the treatment of type 2 diabetes mellitus (T2DM). Currently, six small molecules (Bay 27-9955, MK-0893, MK-3577, LY2409021, PF-06291874, and LGD-6972) have been tested or are undergoing clinical trials, but only the binding site of MK-0893 has been resolved. To predict binding sites for other small molecules, we utilized both the crystal structure of the GCGR and MK-0893 complex and dynamic conformations. We docked five small molecules and selected the best conformation based on binding mode, docking score, and binding free energy. We performed MD simulations to verify the binding mode of the selected small molecules. Moreover, when selecting conformations, results of competitive binding were referred to. MD simulation indicated that Bay 27-9955 exhibits moderate binding stability in Pocket 3. MK-3577, LY2409021, and PF-06291874 exhibited highly stable binding to Pocket 2, consistent with experimental results. However, LY2409021 may also bind to Pocket 5. Additionally, LGD-6972 exhibited relatively stable binding in Pocket 5. We also conducted structural modifications of LGD-6972 based on the results of MD simulations and predicted its analogues’ bioavailability, providing a reference for the study of GCGR small molecules.

## 1. Introduction

Type 2 diabetes mellitus (T2DM) is a metabolic disorder caused by the combination of insulin resistance and insufficient secretion of insulin compensation [1]. Diabetes is accompanied by glucagonemia, and glucagon excess is more critical to the development of diabetes than insulin deficiency [2]. Glucagon is an endogenous peptide containing 29 amino acids that is secreted by islet α cells and primarily targets hepatocytes [3]. Glucagon receptor (GCGR) is a class B G protein-coupled receptor containing three extracellular loops (ECLs) and three intracellular loops that jointly connect seven transmembrane helices [4]. Glucagon mediates severe endogenous hyperglycemia and hyperketonemia in the insulin-deficient state, which is the direct cause of the significantly elevated glucose and ketone levels in patients with severe diabetes [5]. Inhibiting glucagon or blocking glucagon binding to GCGR may be an effective adjunct to conventional anti-hyperglycemic therapy in diabetic patients [6]. Moreover, the specific binding of glucagon and GCGR promotes hepatic glycogenolysis and enhances blood glucose levels to stimulate insulin release [7]. Therefore, the development of agonists or antagonists targeting GCGR can be available in the treatment of T2DM.

The types of drugs targeting GCGR for the treatment of T2DM include small-molecule antagonists, polypeptides [8,9], monoclonal antibodies [10,11,12], and antisense oligonucleotides [13,14]. The GCGR orthosteric site is highly conserved, and orthosteric modulators often interact with highly homologous proteins, which have disadvantages such as low selectivity and serious side effects, hindering the development of highly specific drugs targeting GCGR [15]. Due to the diversity and specificity of allosteric sites, allosteric modulators have the advantages of high selectivity, low side effects, and low toxicity. The development of GCGR small-molecule drugs is currently a hot topic. So far, six small molecules have been tested in clinical trials, including Bay 27-9955 (IC_50_: 110 nM) [16], MK-0893 (IC_50_: 15.7 nM) [17,18], MK-3577 (IC_50_: 13.9 nM) [19], LY2409021 (IC_50_: 1.8 nM) [20,21], PF-06291874 (IC_50_: 112 μM) [22], and LGD-6972 [23,24]. The first five small molecules have been withdrawn from clinical trials due to adverse reactions such as elevated levels of LDL-c and ALT. Only LGD-6972 is still in phase II clinical trials. It is well tolerated and does not cause adverse reactions such as hypoglycemia [23], but its related biochemical information is rarely reported [25].

Up to now, the crystal structures of only two small molecules in complex with GCGR have been resolved, namely MK-0893 [17] and NNC0640 [3], and these two small molecules bind to the same site. Jazayeri et al. [17] not only resolved the binding site of MK-0893 but also selected seven compounds for [^3^H]MK-0893 radioligand binding experiments. The results showed that NNC0640, Cpd-01, Cpd-02, Cpd-03, and Cpd-04, which were chemically similar to MK-0893, could perfectly compete for binding with [^3^H]MK-0893, indicating that these compounds have the same binding site. However, Cpd-05 and Cpd-06, which have different chemical structures from MK-0893, cannot compete for binding with [^3^H]MK-0893, reflecting that these compounds bind to other sites. GCGR small molecules MK-3577 [19], LY2409021 [20,21], and PF-06291874 [22], which have terminated phase II clinical trials, have chemical similarities with MK-0893, and LY2409021 is the Cpd-02 described by Jazayeri [17] that fully competed for binding with [^3^H]MK-0893. Therefore, when we searched for the binding sites of MK-3577 [19], LY2409021 [20,21], and PF-06291874 [22], we gave priority to MK-0893. In addition, we speculate that MK-3577 [19], LY2409021 [20,21], and PF-06291874 [22] may also bind to other sites around MK-0893, and the bound conformation of the compounds occupies part of the MK-0893 site, thus also competing with [^3^H]MK-0893.

Consequently, the main task of this paper is to search for more pockets in order to discover binding sites of GCGR for small molecules (Bay 27-9955 [16], MK-3577 [19], LY2409021 [20,21], PF-06291874 [22], and LGD-6972 [23,24]). We used molecular dynamics (MD) simulations to study the dynamic conformational changes of GCGR in the physiological environment. The shapes and properties of the binding pocket vary in different conformations [26,27]. We extracted representative conformations from MD simulation trajectories of the GCGR apo state as the orthosteric site Pocket 1 and the MK-0893 site as Pocket 2. The four allosteric sites detected by MDpocket [28,29] from MD simulation trajectories of the GCGR/glucagon system were named Pocket 3, Pocket 4, Pocket 5, and Pocket 6. The five GCGR small molecules were docked to six pockets, and conformations with the best binding mode, docking score, and binding free energy were selected to verify the binding mode of the small molecules by MD simulations, so as to find the binding sites of the small molecules in the approximately physiological environment. We also optimized the structure of small molecules based on their binding modes and the characteristics of residues around the pocket and provided new sites for GCGR small molecule researchers to promote the discovery of new drugs targeting GCGR.

## 2. Results

### 2.1. Potential Binding Sites for Small Molecules

We selected five GCGR small molecules that are currently in or have stopped clinical trials and have unknown binding sites to search for their possible binding sites. Among them, the biphenyl compound Bay 27-9955 [16] is the first clinical GCGR small-molecule antagonist, with an IC_50_ of 110 nM. Compounds MK-3577 (IC_50_: 13.9 nM) [19] containing an indole group, LY2409021 (IC_50_: 1.8 nM) [30] containing a biphenyl group, and PF-06291874 (IC_50_: 112 μM) [22] containing a pyrazole group are chemically similar to MK-0893 (IC_50_: 15.7 nM) [17,18] and all contain a 3-(4-methylbenzoylamino) propionic acid group. The compound LGD-6972 [23,24] with a sulfonic acid group is currently the only GCGR small molecule in clinical trials, and its IC_50_ has not been disclosed (Figure 1D). To explore small-molecule binding sites, we used MD simulation to obtain a large number of protein dynamic conformations, thereby collecting more potential binding pockets.

To obtain the conformation of the GCGR orthosteric pocket, we performed GCGR MD simulations of the apo state (Appendix A). We extracted the structure of the last 1 ns as the docking template and named it Pocket 1, which consists of TM1, TM2, TM7, ECL2, as well as the αA Helix of the extracellular domain (ECD) (Figure 1A). So far, only the crystal structures of the complexes of GCGR and two antagonists have been resolved, namely MK-0893 (PDB code: 5EE7 [17]) and NNC0640 (PDB code: 5XEZ [3] and PDB code: 5XF1 [3]). Both antagonists bind to an allosteric site on the outside of the transmembrane helix bundle that extends to the palmitoyl oleoyl phosphatidylcholine (POPC) lipid bilayer between TM6 and TM7, which we have designated as Pocket 2 (Figure 1B). In order to find more allosteric sites, we used MDpocket [28,29] to detect four potential pockets in GCGR/glucagon simulated trajectories, named Pocket 3, Pocket 4, Pocket 5, and Pocket 6 (Figure 1C). Pocket 3 was close to the extracellular region and was located in the pocket formed by glucagon, TM1, TM2, TM3, as well as ECL2. Pocket 4 was located near the intracellular region, within the pockets formed by TM2, TM3, TM6, and TM7. Pocket 5 was located above Pocket 4 and was also composed of TM2, TM3, TM6, and TM7. Pocket 6, adjacent to Pocket 3, was a cavity comprised of glucagon, TM3, TM5, as well as ECL2. The above six pockets were used to predict the binding sites of GCGR for small molecules. In addition, four new pockets discovered using MDpocket have been synchronously subjected to high-throughput virtual screening, and new skeleton active molecules have been discovered, which will be presented in another article.

### 2.2. Verify the Accuracy of the Experimental Method Using MK-0893

The small molecule MK-0893 [17,18] is a GCGR antagonist with a known binding site. We used the crystal structure of the molecule as a positive control, re-docked MK-0893, and performed MD simulations to prove the accuracy and feasibility of the experimental method. The conformation obtained by the molecular docking of MK-0893 almost completely overlapped with the conformation of the crystal structure. The polar group of 3-(4-methylbenzoylamino) propionic acid formed hydrogen bonds with R346^6.37b^, K349^6.40b^, S350^6.41b^, N404^7.61b^, and K405 (numbers in superscript refer to the modified Ballesteros–Weinstein numbering system for class B GPCR [31,32]). Non-polar groups such as biphenyl formed strong hydrophobic interactions with L329^5.61b^ of TM5 and L352^6.43b^ and F345^6.36b^ of TM6 (Figure 2A). This docking conformation was subjected to three MD simulations, and after the system converged, we counted the frequency distribution of small molecule RMSD in the last 20 ns of the three trajectories (Figure 2C). From the frequency profile, MK-0893 mainly had one binding mode. Compared with the crystal structure, the representative conformation of the m-dichlorobenzene and biphenyl group in the MD simulation was slightly deviated. Although it had little effect on the hydrophobic interaction, the deflection was prevented from forming hydrogen bonds with K349^6.40b^ (Figure 2B). In the last 20 ns of the MD simulation, the frequencies of hydrogen bonds between MK-0893 and R346^6.37b^, S350^6.41b^, N404^7.61b^, and K405 were all greater than 0.8 (Figure 2D). This result indicated that the parameters set in the molecular docking and MD simulation are reasonable, and these two methods were used to explore the possible binding sites of the other five GCGR small molecules.

### 2.3. Docking Situation of Five Small Molecules in Six Pockets

Bay 27-9955 had a smaller molecular weight and fewer polar groups, and it was unstable in terms of binding in the large cavity Pocket 1 and Pocket 2. In Pocket 3, Bay 27-9955 formed hydrogen bonds with Q3 and T7 of glucagon, the fluorobenzene group formed π-π stacking with F141^1.39b^ and Y145^1.43b^, and the aliphatic hydrocarbon formed hydrophobic interactions with TM2 and glucagon (Appendix A). And the docking score and binding free energy of Bay 27-9955 in Pocket 3 were the best (Table 1 and Table 2). Studies [17] had shown that small molecules with chemical similarity to MK-0893 were likely to bind at the MK-0893 site, because small molecules MK-3577 and LY2409021 were chemically similar to MK-0893, and their binding modes in Pocket 2 were also similar to MK-0893. Most importantly, LY2409021 had been shown to fully compete for the binding of [^3^H]MK-0893. Therefore, although the docking scores and binding free energies of MK-3577 and LY2409021 in Pocket 2 were not optimal, the binding mode of both was still chosen for MD simulation verification. In addition, MK-3577 had the best docking score and binding free energy in Pocket 4 adjacent to Pocket 2 (Table 1 and Table 2). In Pocket 4, MK-3577 formed hydrogen bonds with N174^2.47b^, H177^2.50b^, Y343^6.34b^, and N404^7.61b^, and it also formed hydrophobic interactions with TM6 and TM7 (Appendix A). LY2409021 had the best docking score and binding free energy in Pocket 5 (Table 1 and Table 2). LY2409021 formed hydrogen bonds with the central hydrogen bond network [33] K187^2.60b^, N238^3.43b^, and Q392^7.49b^, formed π-π stacking with H361^6.52b^, and its biphenyl group formed hydrophobic interactions with TM6 and TM7 (Appendix A). The docking score and binding free energy of PF-06291874 in Pocket 2 were the best (Table 1 and Table 2), and the binding mode was similar to that of MK-0893 (Appendix A). LGD-6972 had the best docking score and binding free energy in Pocket 5 (Table 1 and Table 2). LGD-6972 formed hydrogen bonds with central hydrogen bond networks [33] K187^2.60b^, N238^3.43b^, Y239^3.44b^, and Q392^7.49b^, formed π-π stacking with H361^6.52b^, and biphenyl groups had strong hydrophobic interactions with TM6 and TM7 (Appendix A). In summary, we selected the binding mode of Bay 27-9955 in Pocket 3, MK-3577 in Pocket 2 and Pocket 4, LY2409021 in Pocket 2 and Pocket 5, PF-06291874 in Pocket 2, and LGD-6972 in Pocket 5 for MD simulation to verify the stability of the binding mode.

### 2.4. MD Simulation Was Used to Confirm Binding Modes of Small Molecules

The binding conformation of Bay 27-9955 in Pocket 3 was selected as the initial conformation for MD simulation, and a total of three trajectories were simulated, each of which was run for 180 ns, and the trajectories in the last 20 ns were selected for analysis. From the frequency distribution plot of Bay 27-9955 RMSD values, there were two binding modes for small molecules (Figure 3C). In trajectory Bay-1, the small molecule formed hydrogen bonds with T7 of glucagon, formed π-π stacking with Y145^1.43b^, and also had hydrophobic interactions with I194^2.67b^ and L198^2.71b^ (Figure 3A). In trajectories Bay-2 and Bay-3, small molecules hardly formed hydrogen bonds and mainly formed hydrophobic interactions (Figure 3B). The binding free energy of the three trajectories in the last 20 ns was −25.59 ± 3.09 kcal/mol. The binding free energies of each trajectory were −19.43 kcal/mol, −29.21 kcal/mol, and −28.12 kcal/mol, respectively. The van der Waals energy and polar solvation energy were similar in the three trajectories, and van der Waals energy marked the main contribution. Differently, in trajectory Bay-1, the electrostatic energy was much higher than the other two trajectories, which was also the reason for its overall high binding free energy (Table 3). Energy decomposition showed that the contribution of residues Q3 and T7 on glucagon in trajectory Bay 1 to the binding free energy was positive, mainly because the polar solvation energies of these two residues were high positive values, which were 4.96 kcal/mol and 7.02 kcal/mol, respectively (Figure 3D,E). The energy contribution of Y145^1.43b^ in the two binding modes was also different. In trajectories Bay-2 and Bay-3, Y145^1.43b^ was adjacent to the hydroxyl group of the small molecule, so the electrostatic energy and polar solvation energy were relatively high, resulting in a positive contribution to the binding free energy (Figure 3D,E).

It had been suggested [17] that compounds with chemical similarity to MK-0893 may bind to the same site. Therefore, the binding conformation of MK-3577 in Pocket 2 was used as a template to build the MD simulation system, and a total of three trajectories were simulated, each running for 100 ns, and the trajectories of the last 20 ns were selected for analysis. From the frequency distribution map of MK-3577 RMSD values, although there were two peaks, they were very close to each other (Figure 4C). Analyzing the trajectories, it was found that there was only one binding mode because the RMSD value changed due to the deflection of other groups outside the 3- (4-methylbenzoylamino) propionic acid group (Figure 4A). The hydrogen bonds formed by MK-3577 with residues R346^6.37b^, N404^7.61b^, and K405 were stable in all three simulated trajectories, while the frequency of hydrogen bonds formed with residue S350^6.41b^ fluctuated, especially in the MK3-2 trajectory (Figure 4B). The binding free energy of the three trajectories in the last 20 ns was −29.72 ± 1.82 kcal/mol. The binding free energies of each trajectory were −31.97 kcal/mol, −26.12 kcal/mol, and −31.06 kcal/mol, respectively. Except for the electrostatic energy of the MK-2 trajectory (4.23 kcal/mol), which was a large positive value, van der Waals energy and polar solvation energy were similar for all three trajectories (Table 3). The energy decomposition results of the binding free energy were consistent with the hydrogen bonding frequency, with residues R346^6.37b^, N404^7.61b^, and K405 contributing greatly to the binding of small molecules, and the non-polar aliphatic amino acid L403^7.60b^ also contributed (Figure 4D).

The docking score and binding free energy of MK-3577 in Pocket 4 were the best among the six pockets and were also adjacent to Pocket 2; therefore, MD was used to simulate MK-3577 dynamically changing in Pocket 4. We simulated a total of three trajectories, each of which ran for 180 ns. From the RMSD of MK-3577 over time, it can be seen that small molecules of trajectories MK3-p-1 and MK3-p-2 exhibited significant conformational changes in the first 80 ns, whereas the trajectory MK3-p-3 had always been in a relatively stable state (Appendix A). In trajectories MK3-p-1 and MK3-p-2, the representative conformation of small molecules underwent significant change compared to the initial conformation, with the overall conformation being deviated toward TM6, especially the 3-(4-methylbenzoylamino) propionic acid group more significantly (Appendix A). Consistent with the change trend of RMSD values, the representative conformation of small molecules in trajectory MK3-p-3 was partially similar to the initial conformation, with the most important change being that the 3-(4-methylbenzoylamino) propionic acid group underwent a ‘180°’ deflection, and the carboxyl group was oriented towards TM2 to TM6 (Appendix A). MD simulations of MK-3577 in this pocket all changed its docking binding mode, indicating that this docking binding mode may not exist in the actual physiological state. This result manifested that experiments and calculations should complement each other. Experiments can verify the key conformations and regulatory mechanisms proposed by calculations, and the experimental results can also promote the revision of theoretical computational models.

Studies [17] had shown that LY2409021 competed with MK-0893 in concentration-dependent binding. We used the binding conformation of LY2409021 in Pocket 2 as a template to build the MD system and simulated a total of three trajectories, each running for 100 ns, 100 ns, and 90 ns, respectively. The last 20 ns of trajectories were selected for analysis. From the frequency profile of LY2409021 RMSD, there was only one binding mode for the small molecule (Figure 5C). Analysis of the trajectories also showed that although the binding mode of LY2409021 was the same in each trajectory and formed hydrogen bonds with R346^6.37b^, S350^6.41b^, N404^7.61b^, and K405, the side chain of K405 in trajectory LY-2 ‘warped’ and formed hydrogen bonds with the carboxyl group of LY2409021 (Figure 5A,B). The hydrogen bonds formed by LY2409021 with residues R346^6.37b^, N404^7.61b^, and K405 were all stable over time, while the frequency of hydrogen bonds formed with S350^6.41b^ fluctuated, especially in LY-1 and LY-2 trajectories (Figure 5D). The binding free energy of the three trajectories in the last 20 ns was −47.85 ± 4.09 kcal/mol. The binding free energies of each trajectory were −43.87 kcal/mol, −56.02 kcal/mol, and −43.66 kcal/mol, respectively. Except for the electrostatic energy of the LY-2 trajectory (−18.92 kcal/mol), which was relatively negative, van der Waals energy and polar solvation energy were similar for all three trajectories (Table 3). From the perspective of energy decomposition, the energy contributions of key residues in the three trajectories were not significantly different. Specifically, the contribution of K405 in trajectory LY-2 was significant, which may be related to the side chains of K405 being close to LY2409021 and forming hydrogen bonds with them (Figure 5E). Moreover, the contribution of L403^7.60b^ to binding free energy in trajectory LY-2 was positive, while its contribution to binding energy in trajectories LY-1 and LY-3 was negative. This was because the side chain of L403^7.60b^ in trajectory LY-2 deviated, and its contribution to van der Waals energy was not as significant as in trajectories LY-1 and LY-3 (Figure 5F).

The docking score and binding free energy of LY2409021 in Pocket 5 were the best among the docking results of six pockets, and as a result, we made MD simulations of this binding conformation and submitted a total of three simulation trajectories of 100 ns each. From the frequency profile of LY2409021 RMSD in the last 20 ns of the trajectory, there were two binding modes for the small molecule (Figure 6C). In the LY-p-2 trajectory, LY2409021 was closer to the central region, and the carboxyl group formed hydrogen bonds with K187^2.60b^ of TM2 and N238^3.43b^ of TM3, which were stable throughout the simulation (Figure 6A–D). In the trajectories LY-p-1 and LY-p-3, LY2409021 was shifted toward TM6 and TM7, and the carboxyl group formed hydrogen bonds with H361^6.52b^ of TM6 and Q392^7.49b^ of TM7 (Figure 6B). From the change in hydrogen bonds with time, we clearly observed that LY2409021 gradually transitioned from forming hydrogen bonds with K187^2.60b^ and N238^3.43b^ to forming hydrogen bonds mainly with H361^6.52b^ and Q392^7.49b^. Thus, it can be inferred that LY2409021 slowly moves from being initially close to TM2 and TM3 to TM5 and TM6 (Figure 6D). The binding free energy of the three trajectories in the last 20 ns was −34.78 ± 7.19 kcal/mol. The binding free energies of each trajectory were −29.15 kcal/mol, −49.05 kcal/mol, and −26.13 kcal/mol, respectively. The van der Waals energies and polar solvation energies were similar for all three trajectories, except for the electrostatic energies of LY-p-1 and LY-p-3 (19.68 kcal/mol and 18.01 kcal/mol, respectively), which were relatively large positive values (Table 3). From the energy decomposition, compared with LY-p-1 and LY-p-3, the most special contribution of K187^2.60b^ and N238^3.43b^ to the binding free energy of LY-p-2 trajectory was observed, consistent with the polar role of LY2409021 (Figure 6E). In terms of the binding free energy, the representative binding mode of the LY-p-2 trajectory was the dominant conformation of LY2409021 in Pocket 5.

PF-06291874 was chemically similar to MK-0893 and may bind to the same site [17]. Furthermore, the docking results and binding free energy of PF-06291874 in Pocket 2 were the best among the six pockets, and we used this conformation as the initial conformation for MD simulation. We submitted a total of three trajectories, running 100 ns, 90 ns, and 100 ns, and analyzed the final 20 ns of the trajectories. From the frequency distribution map of PF-06291874 RMSD, there was mainly one binding mode for the small molecule (Figure 7C). Analysis of the trajectory revealed that the peak of the RMSD curve at around 0.4 nm was caused by the deviation of the pyrazole group from TM6, but the polarity interaction of small molecules remained unchanged. Although MD simulation mainly obtained one binding mode of PF-06291874, there were two ways to form hydrogen bonds. The side chain of K405 in trajectory PF-1 ‘curled up’ to form hydrogen bonds with the carboxyl groups of small molecules, while the main chain of K405 in trajectories PF-2 and PF-3 formed hydrogen bonds with small molecules, with the side chain extended ‘straight’ towards the direction of the POPC membrane (Figure 7A,B). In the three trajectories, PF-06291874 had stable hydrogen bonds with R346^6.37b^, N404^7.61b^, and K405 and a less stable hydrogen bond with S350^6.41b^ (Figure 7D). The binding free energy of the three trajectories in the last 20 ns was −31.76 ± 3.48 kcal/mol. The binding free energies of each trajectory were −33.10 kcal/mol, −37.01 kcal/mol, and −25.17 kcal/mol, respectively. The polar solvation energies of the three trajectories were similar, except that the electrostatic energy of trajectory PF-1 (6.86 kcal/mol) was a large positive value and the van der Waals energy of trajectory PF-3 (−24.52 kcal/mol) was not low enough (Table 3). The difference of hydrogen bond formation between K405 and PF-06291874 in the three trajectories can be clearly observed by energy decomposition (Figure 7E).

The docking results and binding free energies of LGD-6972 in Pocket 5 were the best among the six pockets, and we used this conformation as the initial conformation for MD simulations. We submitted three trajectories, each simulated for 130 ns, and the last 20 ns were selected for analysis. From the frequency profile of LGD-6972 RMSD, there was only one binding mode for the small molecule (Figure 8C). In trajectory LGD-1, LGD-6972 formed a stable hydrogen bond with K187^2.60b^ and Q392^7.49b^ and maintained a hydrogen bond with N238^3.43b^ from 100 ns to 130 ns (Figure 8A–D). In trajectory LGD-2, LGD-6972 formed stable hydrogen bonds with H361^6.52b^ and Q392^7.49b^. In trajectory LGD-3, LGD-6972 had a stable hydrogen bond with Q392^7.49b^, while the hydrogen bond with K187^2.60b^ was gradually unstable after 80 ns, and LGD-6972 had an occasional hydrogen bond with H361^6.52b^ from about 40 ns (Figure 8B–D). The binding free energy of the three trajectories in the last 20 ns was −51.37 ± 7.80 kcal/mol. The binding free energies of each trajectory were −63.51 kcal/mol, −53.78 kcal/mol, and −36.81 kcal/mol, respectively. The van der Waals energies and polar solvation energies of three trajectories were similar, except for the electrostatic energies of LGD-2 and LGD-3 (20.29 kcal/mol and 38.32 kcal/mol, respectively), which were relatively large positive values (Table 3). Energy decomposition showed that K187^2.60b^ and N238^3.43b^ in trajectory LGD-1 contribute more to the binding free energy than in trajectory LGD-2 and LGD-3 (Figure 8E). Surprisingly, in the last 20 ns of trajectory LGD-3, LGD-6972 formed hydrogen bonds with H361^6.52b^ at a higher frequency, but its binding free energy was positive. Decomposing the binding free energy of H361^6.52b^ into each energy term, it was found that the high electrostatic energy of H361^6.52b^ was the cause (Figure 8F). In the following, we used LGD-6972 as an example for structural modifications to increase the feasibility of improving the activity or stability of small molecules based on MD simulation.

### 2.5. Structural Modification Based on MD Results

Based on the results of MD simulation, structural modifications are carried out to improve the activity or stability of small molecules, which provides new ideas for GPCR researchers. Here, LGD-6972 was taken as an example to carry out structural modification. For the representative conformation of trajectory LGD-1, the stability of the small molecule in the transmembrane central region was improved by modifying the 2-benzamidoethane-1-sulfonic acid group into the 2-((2-sulfoethyl)carbamoyl)benzoic acid group to increase the polar interaction with H361^6.52b^ (Figure 9A). For the representative conformations of trajectories LGD-2 and LGD-3, the 4-(tert-butyl)-1,1′-biphenyl group was modified to the 4-(tert-butyl)-[1,1′-biphenyl]-3-ol group so that the small molecule formed hydrogen bonds with L350^6.41b^ and L400^7.57b^, increasing the stability of the small molecule between TM6 and TM7 (Figure 9B). LGD-6972 enhanced its binding stability by adding polar groups to enhance its polar interaction with the receptor. In addition, we used SwissADME [34] to predict the bioavailability of LGD-6972 and its analogues, and the results showed that structural modification had little effect on the pharmacokinetic properties of the compounds (Table 4). This structure-based modification method is feasible.

## 3. Discussion

Of the six known GCGR small-molecule antagonists (Bay 27-9955 [16], MK-0893 [17,18], MK-3577 [19], LY2409021 [20,21], PF-06291874 [22], and LGD-6972 [23,24]), only the binding site of MK-0893 had been resolved. In order to predict binding sites for other small molecules, more binding pockets should first be collected. Except the crystal structure of the GCGR and MK-0893 complex (Pocket 2), dynamic conformations were also used. The conformational change in apo GCGR in the physiological condition was simulated, and the conformation of the last 1 ns of the MD simulation trajectory was extracted as a molecular docking template for the orthosteric site (Pocket 1). MDpocket [28,29] was also used to detect potential allosteric sites in MD trajectories of the GCGR/glucagon system, named Pocket 3, Pocket 4, Pocket 5, and Pocket 6. Furthermore, the more detailed process by which MDpocket found new pockets, as well as new molecules discovered by virtual screening, will be presented in a separate article.

POPC phospholipid molecules were added around Pocket 2, and MK-0893 was used as a positive control for re-docking. The docking results were almost completely overlapped with its resolved conformation, and their polar and hydrophobic interactions were basically the same, which verified the accuracy and feasibility of the docking software, docking method, and adding POPC phospholipid molecules. The MD simulation system was built using the docking results of MK-0893 as a template. There was mainly one binding mode for the three trajectories, which was mainly consistent with the crystal structure, indicating the rationality of MD simulation parameters.

Five GCGR antagonists were docked to six binding pockets, and the best conformations with the binding mode, docking score, and binding free energy from the docking results were selected as templates for MD simulations to verify the binding stability of small molecules. Jazayeri et al. [17] found that chemically similar compounds exhibit competitive binding with [^3^H]MK-0893, a factor we also considered when selecting molecular docking results. After molecular docking and MD simulation analysis, it was found that the binding stability of Bay 27-9955 at Pocket 3 was ordinary, and electrostatic energy or hydrophobic interactions can be increased by adding polar or non-polar groups. This is also the reason why compounds MK-0893, MK-3577, LY2409021, and PF-06291874 showed better activity in clinical trials. Although MK-3577 was very stable at Pocket 2, it can also increase its activity by increasing the hydrophobic interaction with TM5 or the polar interaction with T353^6.44b^ or K349^6.40b^ on TM6. For example, LY2409021 and PF-06291874 exhibited better hydrophobic interactions with TM5 and higher activity compared to MK-3577. LY2409021 may also bind to Pocket 5, and its movement in Pocket 5 was mainly due to its unstable polar interaction with TM2 or TM3. Consequently, the binding stability of LY2409021 can be improved through structural modification to increase its polar interaction with TM2 or TM3. The binding of LGD-6972 in Pocket 5 was relatively stable, and structural modifications were carried out based on its two representative conformations. For LGD-1, a carboxyl group was added to the small molecule, forming a hydrogen bond with H361^6.52b^ of TM6. For LGD-2/3, a hydroxyl group was added to the small molecule, forming hydrogen bonds with L350^6.41b^ of TM6 and L400^7.57b^ of TM7. Using representative conformations obtained from MD simulations as a basis for structural modification of compounds can significantly increase the chances of successful modifications and provide new directions for researchers.

## 4. Materials and Methods

### 4.1. Homology Modeling

The full-length inactive conformation of GCGR (PDB code: 5YQZ [35]) was modeled by Modeller [36], and the mutant residues were reverse-mutated to the wild type. The modeled GCGR and glucagon conformations were used as the initial conformations of the MD simulations of the GCGR apo state and GCGR/glucagon complex. Using the crystal structure of small molecule MK-0893 (PDB code: 5EE7 [17]) as a template, a wild-type GCGR conformation was constructed using Modeller [36]. Models with the least root mean square deviation from the crystal structure were selected for templates.

### 4.2. Molecular Dynamics Simulation

To obtain the GCGR orthosteric pocket, we built a simulation system of the GCGR apo state and ran 450 ns trajectories. At the same time, in order to detect more allosteric pockets from MD trajectories, we built a GCGR/glucagon complex simulation system, running a total of 5 trajectories, each trajectory for 1000 ns. Furthermore, in addition to the experimentally verified compounds that may bind at the MK-0893 site in the literature [17], we also performed MD simulations on the conformation with the best docking score and binding free energy in each small molecule docking result to verify the binding mode of small molecules. Three independent simulations were performed for each small-molecule simulation system, and the simulation time and other details are shown in Appendix A.

These models were individually placed in a 93.459 Å × 93.459 Å POPC bilayer and the lipids located within the receptor 1 Å were removed. Simulation systems were solvated with TIP3P water molecules and 0.15M NaCl in a box (93.459 Å × 93.459 Å × 134.825 Å). GROMACS 2020 [37] was used for MD simulations in an isothermal isobaric (NPT) ensemble and periodic boundary condition. The CHARMM36-CAMP force field [38] was applied to proteins, POPC, water molecules, and ions. Small-molecule ligand parameters were generated by the CHARMM Generalized Force Field (CGenFF) [39,40]. For each system, progressive energy minimizations were first performed to mitigate the adverse contact of position constraints. Subsequently, three parallel 50 ns balanced MD runs were performed for each system in the NPT ensemble. During equilibration, temperature and pressure were controlled using the v-rescale method [41] and the Berendsen method [42], respectively. After equilibration, production runs were performed for each simulation system. SETTLE constraints [43] and LINCS constraints [44] were applied to covalently bonded hydrogens involved in water and other molecules, respectively, with a time step set to 2 fs. Electrostatic interactions were calculated using the Particle Mesh Ewald (PME) algorithm [45] with the real-space cutoff of 1.0 nm. The temperature was maintained at 310 K by the v-rescale method [41], and the pressure was kept constant at 1 bar by semi-isotropy coupling to a Parrinello-Rahman barostat [46] with τ_p_ = 2.5 ps and a compressibility of 4.5 × 10^−5^ bar. We used tools and internal scripts from GROMACS 2020 as well as PyMOL (http://www.pymol.org, accessed date 20 July 2024) to analyze simulated data. Analyses were performed using the last 20 ns of the small-molecule simulation system.

### 4.3. MDpocket Detects Pockets in MD Trajectories

MDpocket [28,29] is an open-source tool used to detect potential binding pockets in MD simulation trajectories. Before using MDpocket, it is necessary to extract a PDB file every 100 ps from the processed MD trajectory using GROMACS 2020 [37]. PDB structure ensembles were then detected by MDpocket to output pockets present throughout trajectories, which are able to be observed by the visualization software PyMOL (http://www.pymol.org). This analysis was conducted along five trajectories of the GCGR/glucagon system. MDpocket also allows us to calculate the volume of the selected pockets.

### 4.4. Molecular Docking

To investigate the interaction modes between known active molecules and GCGR, flexible docking was performed using the AutoDock Tools package (version 1.5.6) [47]. The receptor and ligand were optimized to generate the corresponding lower energy 3D conformation and the corresponding ionization state (pH 7.0), respectively. The binding sites found from the dynamic conformation and crystal structure were used as the docking grid center, and the residues around the pocket were set to be flipped. The prepared compounds were then docked to the GCGR pockets and the first 20 conformations of each ligand were exported. The most appropriate binding conformations were selected based on the interaction energy and visual inspection. All results were analyzed and visualized using PyMOL (http://www.pymol.org). To simulate a more realistic physiological environment, POPC phospholipid molecules were added around Pocket 2, Pocket 4, and Pocket 5.

### 4.5. Calculation and Decomposition the Binding Free Energy

Molecular mechanics/Poisson–Boltzmann (Generalized-Born) surface area (MM/PBSA) was one of the most commonly used techniques to calculate the binding free energy. The gmx_MMPBSA tool [48] was used to determine the thermodynamic stability of ligands at binding sites and to examine the contribution of each residue in the binding pocket. The binding free energy of each small-molecule MD simulation system was calculated separately for the last 20 ns (with an interval of 0.5 ns, a total of 40 frames) and energy decomposition was performed. According to the method of calculating the binding free energy of the membrane protein–ligand complex using the CHARMM force field for MD simulation on the official website, files must be prepared. Change the mctrdz in the .in file to the Z value of the center coordinate of the POPC phospholipid molecule, while keeping the other parameters unchanged. Moreover, the binding free energy of each compound in each docking pocket was also calculated [49].

### 4.6. Prediction of Pharmacokinetic Properties

To understand the impact of structural modifications on the pharmacokinetic properties, we use computer models for prediction. SwissADME [34] is a novel web tool that offers free access to a range of rapid and reliable prediction models for physicochemical properties, pharmacokinetics, drug similarity, and pharmaceutical friendliness. We used SwissADME [34] to predict the pharmacokinetic properties of LGD-6972 and its analogs (Compound1 and Compound2).

## Figures and Tables

**Figure 1 ijms-25-08389-f001:**
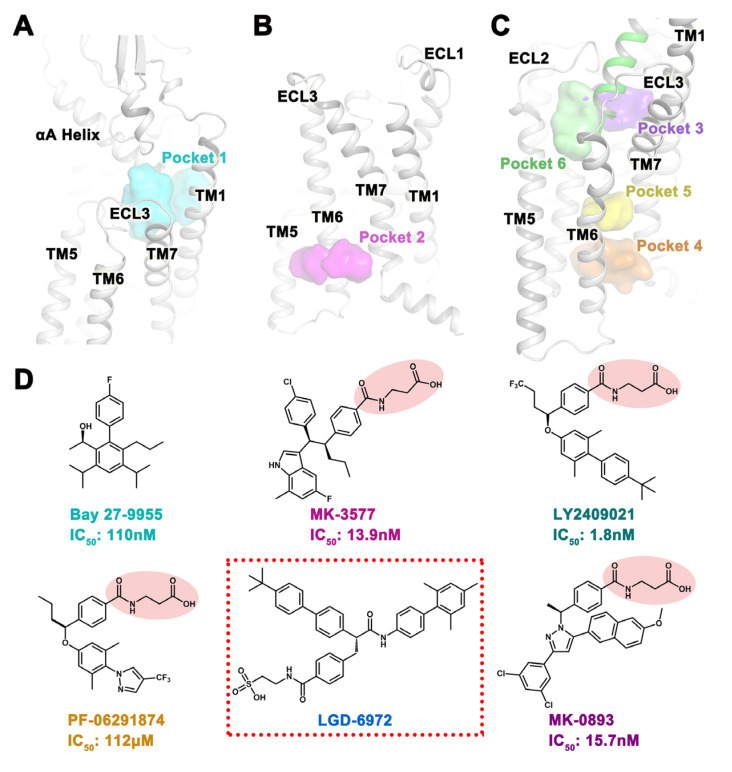
Potential binding pockets and known small molecules of GCGR. (**A**) Orthosteric pocket obtained from GCGR apo state MD simulations. (**B**) Binding site of the small molecule MK-0893. (**C**) MDpocket predicted four allosteric pockets from GCGR/glucagon trajectories. The endogenous ligand glucagon is represented by a green cartoon. Each allosteric site is shown by a different colored surface. (**D**) Compounds with GCGR antagonistic activity that have been reported to be in or have been discontinued in clinical trials. The same groups are represented by pink ellipses. The compound currently undergoing clinical trials is represented by red dashed boxes.

**Figure 2 ijms-25-08389-f002:**
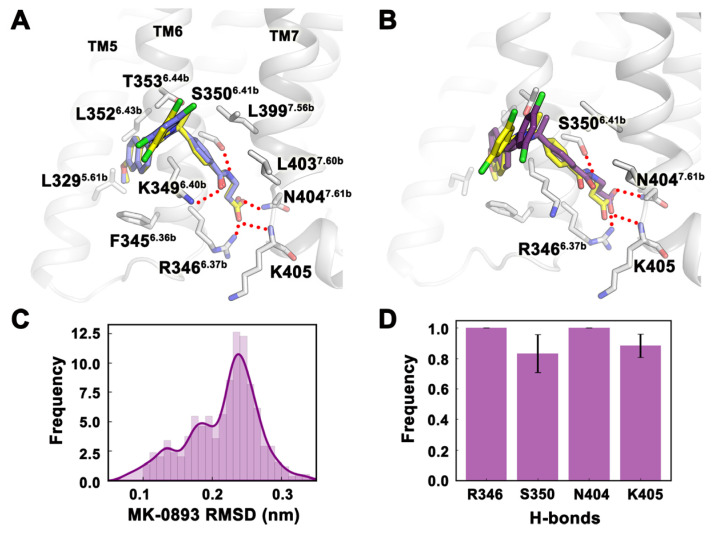
MK-0893 was used to certify the accuracy of the molecular docking and MD simulation methods. (**A**,**B**) The comparison of the representative conformation of the MK-0893 docking (light blue) and MD simulations (purple) with the crystal structure (yellow). Hydrogen bonds are indicated by dashed red lines. (**C**,**D**) RMSD frequency distribution and hydrogen bond frequency profile of small molecule MK-0893 during the last 20 ns of MD simulation (n = 3). Data are shown as mean ± SEM.

**Figure 3 ijms-25-08389-f003:**
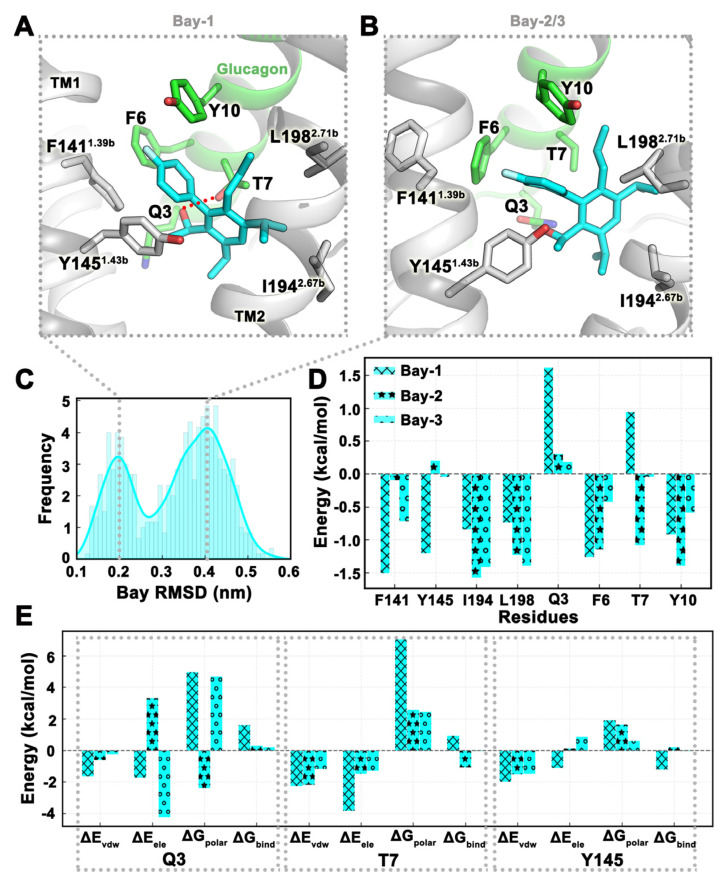
The binding stability of Bay 27-9955 at Pocket 3 revealed in MD simulations. (**A**,**B**) Two representative conformations of Bay 27-9955 (cyan) in MD simulations. (**C**) Frequency curve of Bay 27-9955 RMSD values. (**D**,**E**) The binding free energy of key residues around the pocket as well as the energy decomposition of individual residues. The three trajectories of MD simulation are denoted by ‘x’, ‘*’, and ‘o’, respectively. ΔE_vdw_: van der Waals energy. ΔE_ele_: Electrostatic energy. ΔG_polar_: Polar solvation energy. ΔG_bind_: Total binding free energy. All values are in kcal/mol. All data in this figure are analyzed from the last 20 ns of the MD trajectory.

**Figure 4 ijms-25-08389-f004:**
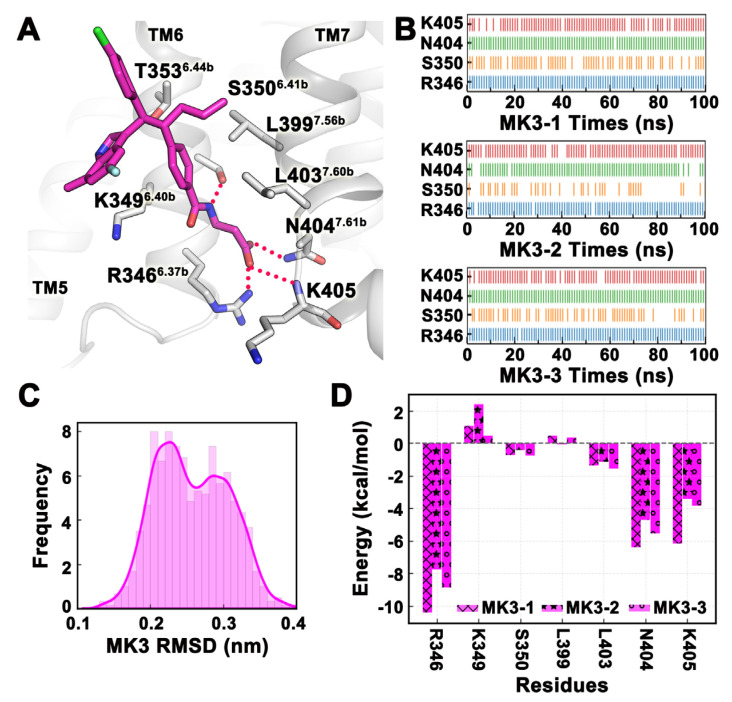
The binding stability of MK-3577 at Pocket 2 revealed in MD simulations. (**A**) The representative conformation of MK-3577 (magenta) in MD simulations. (**B**) Hydrogen bond interactions between MK-3577 and residues over time. (**C**) Frequency curve of MK-3577 RMSD values. (**D**) The binding free energy of key residues around the pocket. The three trajectories of MD simulation are denoted by ‘x’, ‘*’, and ‘o’, respectively. All values are in kcal/mol. All data in this figure are analyzed from the last 20 ns of the MD trajectory.

**Figure 5 ijms-25-08389-f005:**
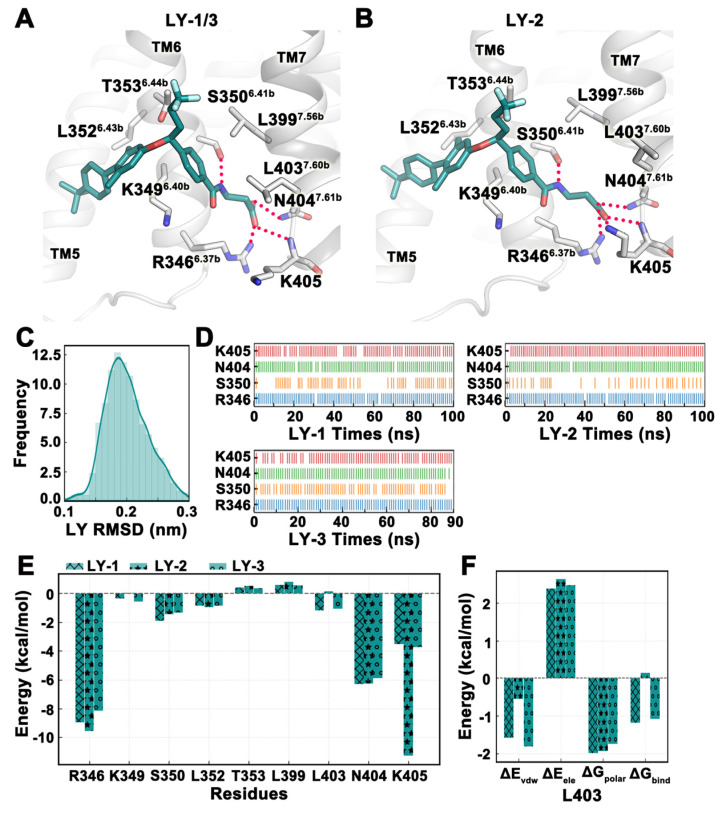
The binding stability of LY2409021 at Pocket 2 revealed in MD simulations. (**A**,**B**) Two representative conformations of LY2409021 (deep teal) in MD simulations. (**C**) Frequency curve of LY2409021 RMSD values. (**D**) Hydrogen bond interactions between LY2409021 and residues over time. (**E**,**F**) The binding free energy of key residues around the pocket as well as the energy decomposition of individual residues. The three trajectories of MD simulation are denoted by ‘x’, ‘*’, and ‘o’, respectively. ΔE_vdw_: van der Waals energy. ΔE_ele_: Electrostatic energy. ΔG_polar_: Polar solvation energy. ΔG_bind_: Total binding free energy. All values are in kcal/mol. All data in this figure are analyzed from the last 20 ns of the MD trajectory.

**Figure 6 ijms-25-08389-f006:**
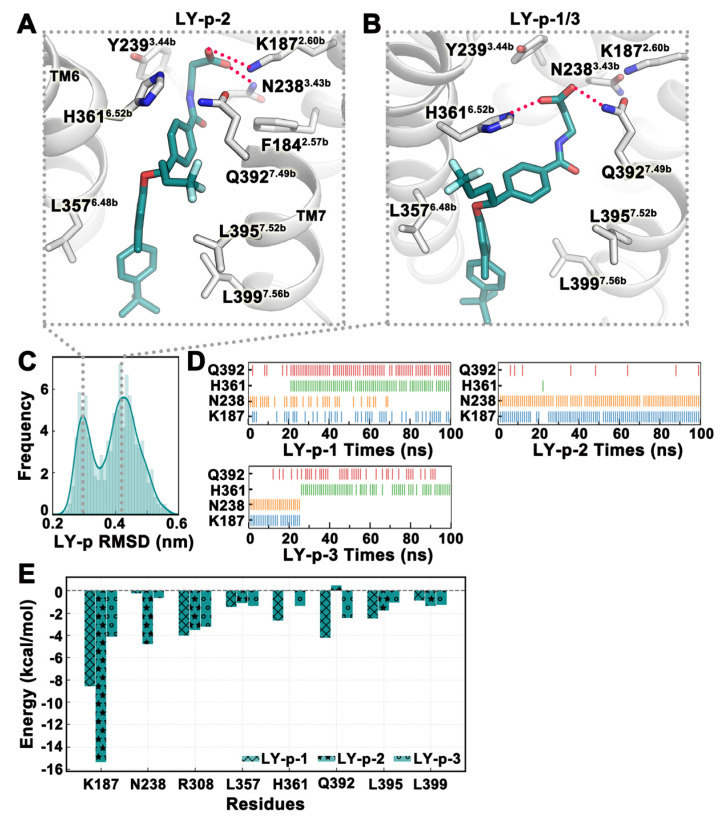
The binding stability of LY2409021 at Pocket 5 revealed in MD simulations. (**A**,**B**) Two representative conformations of LY2409021 (deepteal) in MD simulations. (**C**) Frequency curve of LY2409021 RMSD values. (**D**) Hydrogen bond interactions between LY2409021 and residues over time. (**E**) The binding free energy of key residues around the pocket. The three trajectories of MD simulation are denoted by ‘x’, ‘*’, and ‘o’, respectively. All values are in kcal/mol. All data in this figure are analyzed from the last 20 ns of the MD trajectory.

**Figure 7 ijms-25-08389-f007:**
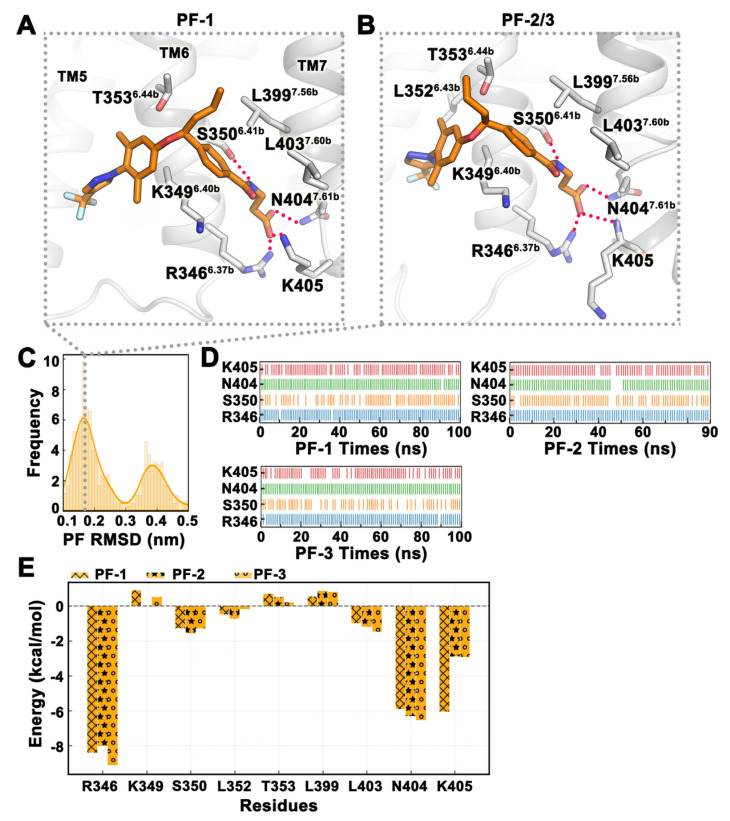
The binding stability of PF-06291874 at Pocket 5 revealed in MD simulations. (**A**,**B**) Two representative conformations of PF-06291874 (orange) in MD simulations. (**C**) Frequency curve of PF-06291874 RMSD values. (**D**) Hydrogen bond interactions between PF-06291874 and residues over time. (**E**) The binding free energy of key residues around the pocket. The three trajectories of MD simulation are denoted by ‘x’, ‘*’, and ‘o’, respectively. All values are in kcal/mol. All data in this figure are analyzed from the last 20 ns of the MD trajectory.

**Figure 8 ijms-25-08389-f008:**
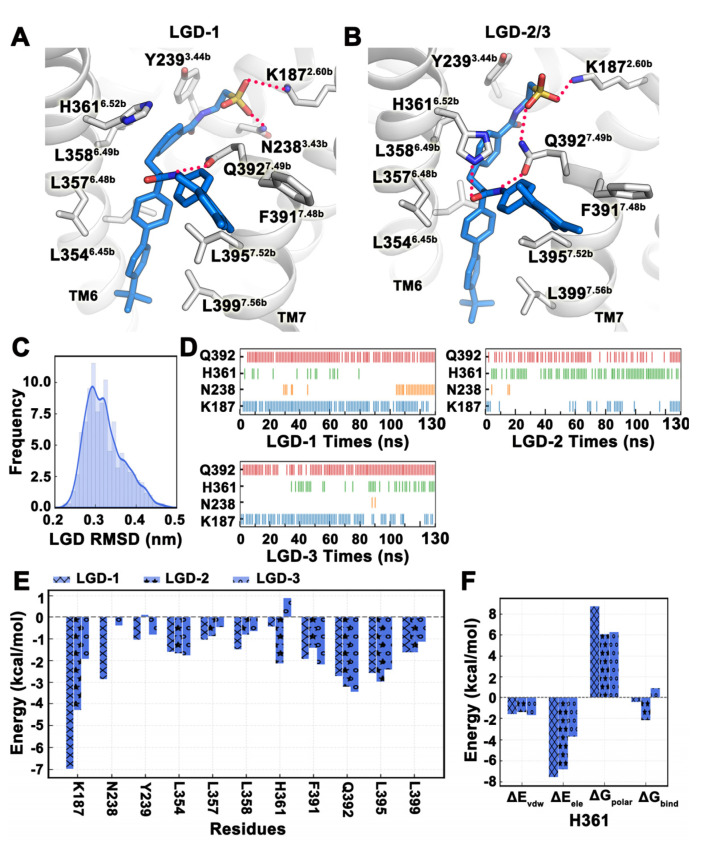
The binding stability of LGD-6972 at Pocket 2 revealed in MD simulations. (**A**,**B**) Two representative conformations of LGD-6972 (marine) in MD simulations. (**C**) Frequency curve of LGD-6972 RMSD values. (**D**) Hydrogen bond interactions between LGD-6972 and residues over time. (**E**,**F**) The binding free energy of key residues around the pocket as well as the energy decomposition of individual residues. The three trajectories of MD simulation are denoted by ‘x’, ‘*’, and ‘o’, respectively. ΔE_vdw_: van der Waals energy. ΔE_ele_: Electrostatic energy. ΔG_polar_: Polar solvation energy. ΔG_bind_: Total binding free energy. All values are in kcal/mol. All data in this figure are analyzed from the last 20 ns of the MD trajectory.

**Figure 9 ijms-25-08389-f009:**
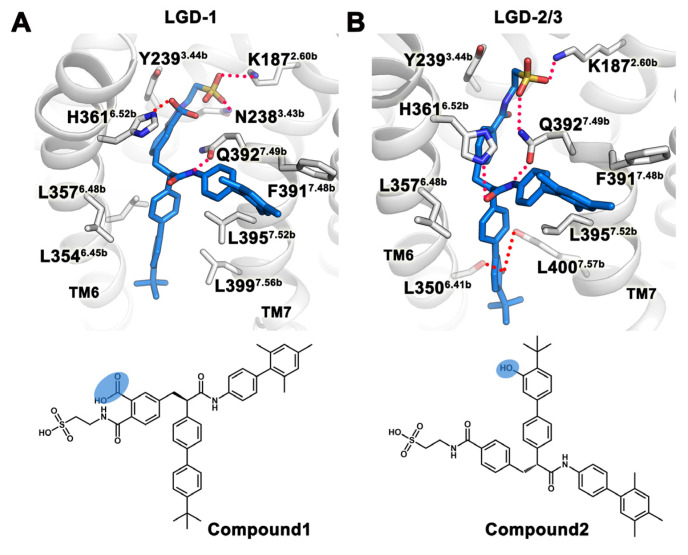
Modified binding modes and 2D structures based on the binding modes of LGD-1 (**A**) and LGD-2/3 (**B**). The modified groups are indicated by blue ellipses.

**Table 1 ijms-25-08389-t001:** Docking score of each small molecule in 6 pockets (kcal/mol).

Molecule	Pocket 1	Pocket 2	Pocket 3	Pocket 4	Pocket 5	Pocket 6
Bay 27-9955			−9.54	−5.12	−6.85	−8.54
MK-3577	−10.65	−8.65	−9.53	−12.69	−12.11	−11.42
LY2409021	−11.22	−12.00	−10.34	−12.55	−16.01	−13.13
PF-06291874	−11.06	−12.05	−9.26	−8.18	−10.75	−11.54
LGD-6972	−13.06	−9.41	−10.86	−10.70	−18.09	−14.29

**Table 2 ijms-25-08389-t002:** Binding free energy of each small molecule in 6 pockets (kcal/mol).

Molecule	Pocket 1	Pocket 2	Pocket 3	Pocket 4	Pocket 5	Pocket 6
Bay 27-9955			−88.59	−47.35	−63.94	−74.75
MK-3577	−96.70	−68.75	−76.62	−105.20	−85.88	−77.54
LY2409021	−95.56	−105.84	−87.94	−111.16	−111.19	−100.29
PF-06291874	−85.33	−85.62	−73.21	−57.33	−81.46	−72.30
LGD-6972	−119.53	−78.44	−99.56	−116.25	−157.91	−142.12

**Table 3 ijms-25-08389-t003:** Calculation results of binding free energy and various energy terms of small molecules in GCGR/small-molecule simulation system (kcal/mol).

System	ΔG_bind_	ΔE_vdw_	ΔE_ele_	ΔG_polar_
Bay-1	−19.43	−44.65	29.89	−4.67
Bay-2	−29.21	−40.75	16.07	−4.53
Bay-3	−28.12	−42.34	18.79	−4.57
Mean ± SD	−25.59 ± 3.09	−42.58 ± 1.13	21.58 ± 4.23	−4.59 ± 0.04
MK-1	−31.97	−25.69	−2.69	−3.60
MK-2	−26.12	−26.66	4.23	−3.69
MK-3	−31.06	−27.62	0.17	−3.62
Mean ± SD	−29.72 ± 1.82	−26.66 ± 0.56	0.57 ± 2.01	−3.64 ± 0.03
LY-1	−43.87	−39.36	0.32	−4.83
LY-2	−56.02	−32.74	−18.92	−4.36
LY-3	−43.66	−40.11	1.24	−4.79
Mean ± SD	−47.85 ± 4.09	−37.40 ± 2.34	−5.79 ± 6.57	−4.66 ± 0.15
LY-p-1	−29.15	−43.42	19.68	−5.41
LY-p-2	−49.05	−45.75	2.58	−5.88
LY-p-3	−26.13	−38.82	18.01	−5.31
Mean ± SD	−34.78 ± 7.19	−42.66 ± 2.04	13.42 ± 5.44	−5.53 ± 0.18
PF-1	−33.10	−35.68	6.86	−4.28
PF-2	−37.01	−34.49	1.79	−4.31
PF-3	−25.17	−24.52	2.65	−3.29
Mean ± SD	−31.76 ± 3.48	−31.56 ± 3.54	3.77 ± 1.57	−3.96 ± 0.34
LGD-1	−63.51	−67.33	11.09	−7.26
LGD-2	−53.78	−66.80	20.29	−7.27
LGD-3	−36.81	−67.83	38.32	−7.29
Mean ± SD	−51.37 ± 7.80	−67.32 ± 0.30	23.23 ± 8.00	−7.27 ± 0.01

ΔG_bind_: Total binding free energy. ΔE_vdw_: van der Waals energy. ΔE_ele_: Electrostatic energy. ΔG_polar_: Polar solvation energy. All data in the table are analyzed from the last 20 ns of the MD trajectory.

**Table 4 ijms-25-08389-t004:** Pharmacokinetic properties of LGD-6972 and its analogues were predicted by SwissADME [34].

Molecule	HBA (≤10)	HBD (≤5)	TPSA (Å^2^)	Synthetic Accessibility (0–10)
LGD-6972	5	3	120.95	5.19
Compound1	7	4	158.25	5.43
Compound2	6	4	141.18	5.24

HBA: hydrogen bond acceptor; HBD: hydrogen bond donor; TPSA: topological polar surface area; Synthetic Accessibility: the estimated easiness of the molecule for synthetic optimization.

## Data Availability

Not applicable.

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
