# Peer review of "The Inferential Binding Sites of GCGR for Small Molecules Using Protein Dynamic Conformations and Crystal Structures"

_ijms, 2024, doi:10.3390/ijms25158389_

Round 1
Reviewer 1 Report
Comments and Suggestions for Authors
A rather good article on inhibitors of glucagon receptors by Wang, Fu, et al.
However, starting with the abstract, the Ms shows that revisions will be needed to improve the language.
There are incomplete sentences:
"Perform molecular docking on 5 small molecules, and the best conformation with the bind-15 ing mode, docking score and binding free energy was selected to perform MD simulation to verify 16 the binding mode of small molecules.
Moreover, when selecting conformations, reference should 17 also be made to experimental results of competitive binding [3H]MK-0893."
In the abstract T2DM is used as an acronym without an explanation.
The Introduction is fine, yet there are details at the level of prepositions
where modifications are vital for understanding the text:
"NNC0640, Cpd-01, Cpd-02, Cpd-03 and Cpd-04, which 60 were chemically similar to MK-0893, could perfectly compete for binding to
[3H]MK-0893, indicating that these compounds have the same binding site"
"NNC0640, Cpd-01, Cpd-02, Cpd-03 and Cpd-04, which 60 were chemically similar to MK-0893, could perfectly compete for binding with
[3H]MK-0893, indicating that these compounds have the same binding site"
and again:
"However, Cpd-05 and Cpd-62 06, which have different chemical structures from MK-0893, cannot compete to bind WITH [3H]MK-0893, reflecting that these compounds bind to other sites."
and:
"that fully competed for binding to [3H]MK-0893"--> "that fully competed for binding with [3H]MK-0893."
"binding sites of GCGR small molecules" --> "binding sites of GCGR for small molecules
Figure 1.
contrast needs to be improved for GCGR and the caption needs to be re-written for clarity.
"MDpocket predicted" --> "MDpocket-predicted": meaning the outlined pockets are predicted using MDpocket.
(e.g. past tense is used for both results and figue colour choices 'each allosteric site was shown'
--> 'each allosteric site is shown')
Table 2. --> Free energies of binding - could the authors also report errors on these numbers from repeated simulations?
Figure 3. same question
The discussion and conclusions are generally sound, with the same remark for English, e.g.:
"Taking representative conformations by 450 MD simulations as an example for compounds structural modification, could greatly imncrease the possibility of successful modification and provide new directions for researchers."
--> unclear
Materials and methods --> I do appreciate the authors' use of Modeller, but isn't AlphaFold superior at this point?
If in this case it is not, could the authors explain why?
The paper needs to be revised by a native English-speaking colleague, this is vital for the described research.
By this I do not mean that most of the text is badly-written, just that there are essential details that need the attention of a proofreader.
It would be better if the authors tried less to be concise. If they chose descriptive language, the results would gain in clarity. Seeking concision in English comes with lots of semantic traps, the sense of phrases
can be altered completely, which is deadly for a research article.
Author Response
Response to Reviewer 1 Comments
- Summary
Thank you very much for taking the time to review this manuscript. Please find the detailed responses below and the corresponding revisions highlighted changes in the re-submitted files.
- Point-by-point response to Comments and Suggestions for Authors
Comments 1: However, starting with the abstract, the Ms shows that revisions will be needed to improve the language. There are incomplete sentences: "Perform molecular docking on 5 small molecules, and the best conformation with the bind-15 ing mode, docking score and binding free energy was selected to perform MD simulation to verify 16 the binding mode of small molecules. Moreover, when selecting conformations, reference should 17 also be made to experimental results of competitive binding [3H]MK-0893." In the abstract T2DM is used as an acronym without an explanation.
Response 1: Thank you for pointing this out. We have revised the incomplete sentences: “Dock five small molecules, select the best conformation with 15 binding mode, docking score and binding free energy to perform MD simulation to verify the bind-16 ing mode of small molecules. Moreover, when selecting conformations, the experimental results of 17 competitive binding [3H]MK-0893 should also be referred to.” And we have supplemented the full name of T2DM in the abstract.
Comments 2: The Introduction is fine, yet there are details at the level of prepositions where modifications are vital for understanding the text: "NNC0640, Cpd-01, Cpd-02, Cpd-03 and Cpd-04, which 60 were chemically similar to MK-0893, could perfectly compete for binding to [3H]MK-0893, indicating that these compounds have the same binding site", "NNC0640, Cpd-01, Cpd-02, Cpd-03 and Cpd-04, which 60 were chemically similar to MK-0893, could perfectly compete for binding with [3H]MK-0893, indicating that these compounds have the same binding site", and again: "However, Cpd-05 and Cpd-62 06, which have different chemical structures from MK-0893, cannot compete to bind WITH [3H]MK-0893, reflecting that these compounds bind to other sites." And: "that fully competed for binding to [3H]MK-0893"--> "that fully competed for binding with [3H]MK-0893." "binding sites of GCGR small molecules" --> "binding sites of GCGR for small molecules
Response 2: We are most grateful for the modifications you suggested, which are crucial for understanding the text. The modifications to the prepositions have been completed as required.
Comments 3: Figure 1. contrast needs to be improved for GCGR and the caption needs to be re-written for clarity. "MDpocket predicted" --> "MDpocket-predicted": meaning the outlined pockets are predicted using MDpocket. (e.g. past tense is used for both results and figue colour choices 'each allosteric site was shown' --> 'each allosteric site is shown')
Response 3: Thank you for pointing this out. We have made modifications as requested.
Comments 4: Table 2. --> Free energies of binding - could the authors also report errors on these numbers from repeated simulations? Figure 3. same question
Response 4: Thank you very much for raising this issue. Table 2 shows the best binding free energy of each small molecule in each pocket, so only one value is listed. Table 3 details the binding free energies and various energy terms from three replicate simulations. Table 3 and Figure 3 aim to show the similarities and differences in binding free energy among different simulation trajectories of the same system, and to identify which energy terms or residues are responsible for these differences. Therefore, the errors between the simulations were not calculated.
Comments 5: The discussion and conclusions are generally sound, with the same remark for English, e.g.: "Taking representative conformations by 450 MD simulations as an example for compounds structural modification, could greatly imncrease the possibility of successful modification and provide new directions for researchers." --> unclear
Response 5: Thank you for pointing out the issue. We agree with you very much and have made revisions ‘Using representative 460 conformations obtained from MD simulations as a basis for structural modification of 461 compounds can significantly increase the chances of successful modifications and provide 462 new directions for researchers.’
Comments 6: Materials and methods --> I do appreciate the authors' use of Modeller, but isn't AlphaFold superior at this point? If in this case it is not, could the authors explain why?
Response 6: Thank you very much for raising this question, which has given us the opportunity to discuss with you. AlphaFold has indeed demonstrated superior performance in protein structure prediction compared to traditional methods like Modeller in many cases. Since this paper only requires modeling of already resolved crystal structures and computational resources are limited, using Modeller will suffice to meet the needs.
- Response to Comments on the Quality of English Language
Point 1: The paper needs to be revised by a native English-speaking colleague, this is vital for the described research. By this I do not mean that most of the text is badly-written, just that there are essential details that need the attention of a proofreader.
Response 1: Thank you for pointing this out. The article has been revised by a native English-speaking colleague.
Point 2: It would be better if the authors tried less to be concise. If they chose descriptive language, the results would gain in clarity. Seeking concision in English comes with lots of semantic traps, the sense of phrases can be altered completely, which is deadly for a research article.
Response 2: Thank you for pointing this out. We have taken the feedback into account and have supplemented and refined the sentences that did not need to be concise.
Reviewer 2 Report
Comments and Suggestions for Authors
The authors have done a rigorous job of identifying binding sites of 6 small molecules within the human glucagon receptor in the presence of glucagon through molecular dynamics simulation. The results are presented in a clear fashion, and I believe the predicted binding sites can be readily tested with structural studies. Only some clarifications are required before publication of this manuscript:
(1) The acronyms of ECL and ECD are not defined anywhere (lines 105-106). Same for POPC in line 110. It would be nice with these ECL (extracellular loops) can be labeled in Figure 1.
(2) The authors should briefly explain the nomenclature of superscripts that accompany transmembrane residues.
(3) Why are polar solvation energies represented in terms of changes in free energies, while van der Waals and electrostatic energies are calculated in terms of changes in internal energies?
Author Response
Response to Reviewer 2 Comments
- Summary
Thank you very much for taking the time to review this manuscript. Please find the detailed responses below and the corresponding revisions highlighted changes in the re-submitted files.
- Point-by-point response to Comments and Suggestions for Authors
Comments 1: The acronyms of ECL and ECD are not defined anywhere (lines 105-106). Same for POPC in line 110. It would be nice with these ECL (extracellular loops) can be labeled in Figure 1.
Response 1: Thank you for pointing this out. We have supplemented the full name of ECLs, ECD and POPC in lines 31-32, 105 and 109, respectively. Also, we have marked ECLs in Figure 1.
Comments 2: The authors should briefly explain the nomenclature of superscripts that accompany transmembrane residues.
Response 2: Thank you for pointing this out. We have explained the naming method for residue superscripts in lines 137-138, namely “numbers in superscript refer to the modified Ballesteros–137 Weinstein numbering system for class B GPCR [31, 32]”.
Comments 3: Why are polar solvation energies represented in terms of changes in free energies, while van der Waals and electrostatic energies are calculated in terms of changes in internal energies?
Response 3: Thank you very much for raising this question, which has given us the opportunity to discuss with you. We calculated the binding free energy using Molecular mechanics/Poisson-Boltzmann (Generalized-Born) surface area (MM/PBSA). In this method, the total binding free energy is divided into two parts: molecular mechanical terms (binding free energy in vacuum) and solvation energies. Molecular mechanical terms can be divided into bond energy and non-bond energy. The bond energy includes bonds, angles and dihedrals, and the non-bond energy includes van der Waals energy and electrostatic energy. Because there is little conformational change before and after receptor-ligand binding, van der Waals and electrostatic energies are mainly considered. Solvation energies can be divided into two parts: polar solvation energy and non-polar solvation energy. Because small molecules studied in this article are located in the transmembrane region, polar solvation energies are mainly considered. Therefore, van der Waals energy, electrostatic energy and polar solvation energy are used to represent the total binding free energy in this paper.
Reviewer 3 Report
Comments and Suggestions for Authors
Wang et al. report a structure-interaction study of six low molecular weight ligands as GCGR drugs. Using a combination of MD simulations, molecular docking and free binding energy predictions, they estimated the binding modes of selected ligands in GCCR. The work is well prepared, the chosen methodology corresponds to the standards, it avoids ambiguities and the conclusions correspond to the obtained results. In principle, it is a work that could attract a wider audience. Although it is only a theoretical work, it could serve as a starting point for further experimental research. In principle, it is possible to publish the work as it is. But I would recommend few minor changes that might improve a bit the paper.
1) In molecular docking, authors used first 10 conformations. Is that enough? Can you show for one selected ligand some extended sampling set? E.g. that conformation 20 is not better than the first ten?
2) Do you think that the presented results are significantly dependent on the force field used? Can a different force field significantly change the observed and described binding sites?
3) chapter 2.5 - modification of LGD; I would supplement this section with ADME/Tox predictions for the original LGD and its analogues to show whether the modification is/is not at the expense of pharmacokinetic properties.
4) page 5, first paragraph; there is written: "Although the docking score and binding free energy of MK-3577 and LYS2409021 in Pocket 2 were not optimal, the binding mode of both was still selected for MD simulation verification." Why? I think, some explanation is needed here.
After taking into account the above-mentioned minor issues, I will gladly recommend the article for publication in IJMS.
Author Response
Response to Reviewer 3 Comments
- Summary
Thank you very much for taking the time to review this manuscript. Please find the detailed responses below and the corresponding revisions highlighted changes in the re-submitted files.
- Point-by-point response to Comments and Suggestions for Authors
Comments 1: In molecular docking, authors used first 10 conformations. Is that enough? Can you show for one selected ligand some extended sampling set? E.g. that conformation 20 is not better than the first ten?
Response 1: Thank you for pointing this out. We agree with you that a larger sample size is beneficial to obtain more accurate molecular docking results. Because the docking results of small molecules studied in this article are very stable, only the first 10 conformations were used. In order to fully sampling, we reanalyze the first 20 conformations and the results show that the optimal docking score and binding free energy of each small molecule in each pocket are the same as the previous results.
Comments 2: Do you think that the presented results are significantly dependent on the force field used? Can a different force field significantly change the observed and described binding sites?
Response 2: Thank you very much for raising this question, which has given us the opportunity to discuss with you. We believe that the presented results are the conformation with the highest frequency of occurrence, which should not depend on the force field used. Under different force fields, binding sites may have a little difference, but will not significantly changes.
Comments 3: chapter 2.5 - modification of LGD; I would supplement this section with ADME/Tox predictions for the original LGD and its analogues to show whether the modification is/is not at the expense of pharmacokinetic properties.
Response 3: Thank you for pointing this out. We agree with this comment. “We used SwissADME [34] to predict the bioavailability of LGD-6972 and its analogues, and the results showed that structural modification had little effect on the pharmacokinetic properties of the compounds (Table 4). This structure-based modification method is feasible.”
Comments 4: page 5, first paragraph; there is written: "Although the docking score and binding free energy of MK-3577 and LY2409021 in Pocket 2 were not optimal, the binding mode of both was still selected for MD simulation verification." Why? I think, some explanation is needed here.
Response 4: Thank you for pointing this out. Perhaps it was due to my unclear description, and I have already made modifications to these sentences. “Because small molecules MK-3577 and LY2409021 were chemically similar to MK-0893, and their binding modes in Pocket 2 were also similar to MK-0893. Most importantly, LY2409021 had been shown to fully competed for the binding of [3H]MK-0893. Therefore, although the docking scores and binding free energies of MK-3577 and LY2409021 in Pocket 2 were not optimal, the binding mode of both was still chosen for MD simulation verification.”